

# Role of reactive oxygen species and isoflavonoids in soybean resistance to the attack of the southern green stink bug

Ivana Sabljic[1,2], Jesica A. Barneto[1], Karina B. Balestrasse[1],
Jorge A. Zavala[1] and Eduardo A. Pagano[1]

[1] Instituto de Investigaciones en Biociencias Agrícolas y Ambientales-INBA, Facultad de Agronomía, Universidad de Buenos Aires, Consejo Nacional de Investigaciones Científicas y Técnicas, Ciudad Autónoma de Buenos Aires, Argentina
[2] GDM, Chacabuco, Buenos Aires, Argentina

## ABSTRACT

Southern green stink bugs (*Nezara viridula* L.) are one of the major pests in many soybean producing areas. They cause a decrease in yield and affect seed quality by reducing viability and vigor. Alterations have been reported in the oxidative response and in the secondary metabolites in different plant species due to insect damage. However, there is little information available on soybean-stink bug interactions. In this study we compare the response of undamaged and damaged seeds by *Nezara viridula* in two soybean cultivars, IAC-100 (resistant) and Davis (susceptible), grown under greenhouse conditions. Pod hardness, $H_2O_2$ generation, enzyme activities in guaiacol peroxidase (GPOX), catalase (CAT) and superoxide dismutase (SOD) as well as lipoxygenase expression and isoflavonoid production were quantified. Our results showed a greater resistance of IAC-100 to pod penetration, a decrease in peroxide content after stink bug attack, and higher GPOX, CAT and SOD activities in seeds due to the genotype and to the genotype-interaction with the herbivory treatment. Induction of *LOX* expression in both cultivars and higher production of isoflavonoids in IAC-100 were also detected. It was then concluded that the herbivory stink bug induces pathways related to oxidative stress and to the secondary metabolites in developing seeds of soybean and that differences between cultivars hold promise for a plant breeding program.

# INTRODUCTION

The stink bug complex is one of the most detrimental pests that affects soybean (*Glycine max* L.) yields in many countries. In South America, the main species that impair soybean yields are *Nezara viridula*, *Piezodorus guildinii* and *Euschistus heros*. They adversely affect seed vigor and may even kill the embryo and stop germination. Brazil and Argentina are two of the most important producers of this legume and damage caused by the mentioned species can result in significant economic losses (*Schaefer & Panizzi, 2000*). Soybean crops destined for seed production are less tolerant to stink bug damage

Corresponding author
Eduardo A. Pagano,
epagano@agro.uba.ar

than soybean crops used for animal feed (*Jensen & Newsom, 1972*). While stink bugs can be controlled by spraying pesticides, there are environmental and economic issues concerning the frequent use of these products. Biological control can be mentioned as an alternative method to fight against this complex of insects. For example, the well-studied egg-parasitoid *Trissolcus basalis* (Wollaston) has been successfully used by farmers in Brazil (*Corrêa-Ferreira & Moscardi, 1996*). However, soybean host plant resistance is an interesting strategy for managing plant breeding programs (*Boerma & Walker, 2005*), and it is necessary to first understand the mechanisms related to stink bug resistance in soybean breeding lines.

The soybean cultivar IAC-100 has been defined as resistant against stink bug attack (*Campos et al., 2010*; *McPherson, Buss & Roberts, 2007*; *Piubelli et al., 2003a*; *De Souza et al., 2014*; *Michereff et al., 2011*) and is used in breeding programs as a source of resistance (*McPherson, Buss & Roberts, 2007*). This variety was developed by the Agronomic Institute of Campinas in Sao Paulo (Brazil) by crossing breeding lines IAC 78-2318 and IAC-12 (*Priolli et al., 2002*). Studies on this genotype showed that stink bug attack caused less seed damage and less seed weight loss than other genotypes (*Campos et al., 2010*). Previous studies found lower adult fresh weight of *Nezara viridula* when fed on "IAC-100" and less lipids in females than in those fed on the other genotypes (*Piubelli et al., 2003b*). However, the mechanisms through which soybean reacts to stink bug attack are still unknown.

Plants respond to herbivore injury through several direct and indirect morphological, biochemical and molecular mechanisms that help them avoid herbivore attack or affect its performance (*Fürstenberg-Hägg, Zagrobelny & Bak, 2013*). Direct defense mechanisms affect insect performance and feeding behavior, while indirect defense mechanisms can attract the natural enemies of the herbivores and thus reduce plant loss (*War et al., 2012*; *Freeman & Beattie, 2008*). Insect herbivory induces early responses such us the oxidative burst (production of reactive oxygen species-ROS), the expression of defense-related genes, and late responses such as callose deposition and accumulation of proteinase inhibitors (*Savatin et al., 2014*). Infestation of *Vicia faba* by *N. viridula* significantly stimulates the production of $H_2O_2$ (*Ederli et al., 2017*). Moreover, in poplar the concentrations of $H_2O_2$ and malondialdehyde and the activities of ROS-scavenging enzymes, such as superoxide dismutase (SOD), catalase (CAT) and peroxidase (POD), were enhanced by herbivore wounding, suggesting that they are associated with insect resistance. The above-mentioned enzymes are related to plant signaling, synthesis of defense compounds, and to oxidative stress tolerance (*Bi & Felton, 1995*; *Zhang, Hua & Zhang, 2008*; *Mai et al., 2013*; *Zebelo & Maffei, 2014*).

After insect attack the activation of defense genes that stimulate the production of the antiherbivory compounds is induced due to internal signals such as calcium ion flow, a phosphorylation cascade and responses mediated by hormones: jasmonic acid (JA) and ethylene (ET) (*Wasternack, 2007*; *Howe & Jander, 2008*; *Browse, 2009*). JA and its methyl ester, methyl jasmonate (MJ), participate in the activation of plant defense mechanisms as signaling compounds in processes related to production of various secondary metabolites (i.e., terpenoids, phenylpropanoids and alkaloids) (*Wang et al.,*

2015; *Misra et al., 2014*). In the case of soybean, isoflavones are the main chemical defense compounds against insects because they affect the performance and survival of the herbivores (*Piubelli et al., 2003a*). It is well known that MJ and JA induce the accumulation of proteinase inhibitors as direct defense mechanisms against herbivorous insects (*Farmer & Ryan, 1990*; *Farmer, Johnson & Ryan, 1992*). Studies on soybean analyzed the effects of jasmonic acid induction and showed that the soybean looper (*Chrysodeixis includens* Walker) selects plant leaves that have not been induced by JA *Accamando & Cronin (2012)*. In addition, soybean crops grown under conditions of high concentrations of ambient $CO_2$ were unable to express genes related to the JA synthesis, which made them highly susceptible to insect attack (*Zavala et al., 2008*). The synthesis of jasmonates and many other oxilipins is initiated by lipoxygenases (LOXs), which catalyze dioxygenation of polyunsaturated fatty acids (reviewed by *Blée (2002)*, *Feussner & Wasternack (2002)*, *Howe & Schilmiller (2002)* and *Wasternack (2007)*). LOX activity was higher in soybean plants that were less vulnerable to the attack of *A. gemmatalis* when compared to controls (*Fortunato et al., 2007*). Since three lipoxygenases, Lox1, Lox2 and Lox3 were detected in soybean seeds (*Axelrod, Cheesbrough & Laakso, 1981*), it is possible that seed lipoxygenases regulate defenses against stink bug attack. As a first step to develop a resistant cultivar, it is necessary to identify the sources of resistance for any breeding program (*De Morais & Pinheiro, 2012*). Therefore, it is important to determine the different characteristics of the diverse soybean cultivars as regards plant defense against insect attack.

The aim of this study is to identify and quantify physical and biochemical soybean defense mechanisms against the southern green stink bug, *Nezara viridula* attack. Our hypothesis is that herbivory by stink bugs lead to biochemical changes and that there are differences between cultivars at both biochemical and physical level. In this study we have compared soybean seed response to stink bug damage of two cultivars, IAC-100 (resistant) and Davis (susceptible; *Orr, Boethel & Jones, 1985*; *Kester, Smith & Gilman, 1984*). Our results show that there are differences between genotypes as regards the oxidative stress response and isoflavonoid production after to stink bugs attack. A better understanding of the resistance pathways can help find ways to improve varieties.

## MATERIALS AND METHODS

### Insect and plant material

In order to determine soybean resistance to stink bug damage (*Nezara viridula* L.) in developing seeds, two soybean (*Glycine max* L.) cultivars, IAC-100 and Davis, were grown under greenhouse conditions and stink bug adults were placed and allowed to feed on pods. Plants were grown with 16:8 L:D light regime at 26 ± 3 °C, in pots with 3:1:1:1 soil: peat:sand:perlite mixture. The colony of stink bugs consisted of field-collected individuals from the city of Chacabuco (34°38′00″S 60°28′00″O) in the province of Buenos Aires (Argentina). The insects were reared at our laboratory (School of Agronomy, University of Buenos Aires, Argentina) on a diet based on hydrated soybean seeds (cv. Williams) and peeled sunflower seeds, and were then provided water with ascorbic acid (0.5 w/v%) (*Giacometti et al., 2016*).

## Experimental design

Four experiments were carried out under the same greenhouse conditions in a completely randomized design. In the first experiment, seventeen replications of each cultivar were used, pod hardness was determined by collecting pods at the R6 soybean development stage (*Fehr et al., 1971*) and by using a texturometer. In order to determine oxidative stress response of developing seeds to stink bug damage in the second experiment, a single adult stink bug was bagged in on one soybean pod of plants ($n = 5$) at the R6 development stage (*Fehr et al., 1971*). Pods of control plants were covered with tulle bags and collected 72 h after stink bugs started feeding. Soon after, they were fast frozen in liquid nitrogen and stored at −80 °C until analysis. In addition, attacked and unattacked pods (one per plant) were collected for $H_2O_2$ detection in developing seeds. Five replications of each cultivar were used for stink bug infestation with an equal number of control plants. In the third experiment undamaged and damaged seeds by stink bugs were collected to determine isoflavonoid accumulation after 72-h herbivory. Finally, a fourth experiment was performed to determine the LOX expression level in damaged and undamaged seeds, and pods were collected 24-h after herbivory and were fast frozen in liquid nitrogen.

## Hardness determination on pods

Hardness, the peak force required to penetrate a soybean pod, was measured with an INSTRON 4442 tensile tester calibrated to penetrate 12 mm at a rate of 2.0 mm sec$^{-1}$ with a needle. The tests were performed on both soybean cultivars at the R6 phenological stage and the needle was inserted in pods at the second seed position. Results were expressed as gram-force per square centimeter (gf cm$^{-2}$) and represented the maximum force required to penetrate a pod with the tip needle in order to simulate the situation in which the sting bugs insert the stylus (*De Santana Souza et al., 2013*).

## Oxidative stress response

Extracts for determination of catalase (CAT), superoxide dismutase (SOD) and Guaiacol peroxidase (GPOX) activities were prepared from 0.3 g of soybean seeds homogenized under ice-cold conditions in 3 mL of extraction buffer, containing 50 mM phosphate buffer (pH 7.4), 1 m MEDTA, 1 g PVP and 0.5% (v/v) Triton X-100 at 4 °C.
The homogenates were centrifuged at 10,000×g for 20 minutes and the supernatant fraction was used for the assays. Total protein content was determined by following the Bradford method for protein quantification (*Bradford, 1976*).

CAT activity was determined according to a modified protocol of *Chance, Sies & Boveris (1979)*. The reaction was conducted in a quartz cuvette by measuring the decrease in absorption at 240 nm in a reaction medium containing 150 μL of enzyme extract, 50 mM potassium phosphate buffer (pH 7.2) and 2 mM $H_2O_2$ for 2 min. CAT activity was determined using the molar absorptivity of $H_2O_2$ at 240 nm and expressed as μmoles mg protein$^{-1}$ min$^{-1}$.

SOD activity was assayed by the inhibition of the photochemical reduction of NBT (*Becana et al., 1986*). The reaction mixture consisted of 50, 100 and 200 μL of enzyme extract and 3.5 mL O$_2$•− generating solution which contained 14.3 mM methionine,

**Table 1  LOX1, LOX2 and ELF1b primer sequences.**

| | | | |
|---|---|---|---|
| *LOX1* | Forward: | 5′ CTGGTGTAAATCCCTGCGTAA 3′ | *Chen et al. (2012)* |
| | Reverse: | 5′ TACCAAGTGCCTCGTCCATT 3′ | |
| *LOX2* | Forward: | 5′ AGATGGTTGCGGGTGTAAAT 3′ | *Chen et al. (2012)* |
| | Reverse: | 5′ GGGCATCTGCTGTTATCTTAC 3′ | |
| *ELF1b* | Forward: | 5′-GTTGAAAAGCCAGGGGACA-3′ | *Jian et al. (2008)* |
| | Reverse: | 5′-TCTTACCCCTTGAGCGTGG-3′ | |

82.5 μM NBT and 2.2 μM riboflavin. Extracts were brought to a final volume of 0.3 mL with 50 mM K-phosphate (pH 7.8) and 0.1 mM Na$_2$EDTA.

Sample tubes were shaken and placed in front of fluorescent lamps during 10 min. The reduction in NBT was followed by reading absorbance at 560 nm. Blanks and controls were run in the same way but without illumination and enzyme, respectively. One unit of SOD was defined as the amount of enzyme which produced a 50% inhibition of NBT reduction under the assay conditions.

GPOX activity was determined according to the protocol devised by *Shannon, Kay & Lew (1966)* which consists in measuring the increase in absorption at 470 nm due to the formation of tetraguaiacol at 30 °C in a reaction extract, 50 mM buffer K-phosphate 50 mM, pH 7, 0.1 mM EDTA, 10 mM guaiacol and 10 mM H$_2$O$_2$. One unit of POD forms 1 μmol of guaiacol oxidized per min under assay conditions.

## Detection of H$_2$O$_2$

To visually analyze H$_2$O$_2$ generation, stink bug damaged seeds and their respective controls were excised and immersed in a 1% solution of 3,3-Diaminobenzidine (DAB) for 24 h under light at 25 °C (*Zilli et al., 2008*). DAB generates a reddish-brown compound that could be detected at the site of H$_2$O$_2$ formation in the presence of endogenous peroxidase activity. After staining, formation of brown precipitates was documented by photography.

## Lipoxygenase expression

The expression level of Lipoxygenase 1 (*LOX1*) and Lipoxygenase 2 (*LOX2*) genes in soybean seeds was determined. Briefly, total RNA was isolated using the RNeasy Plant Mini kit (QIAGEN Inc., Valencia, CA, USA) according to the manufacturer's protocol. RNA concentration was determined with a fluorometer Qubit™ (Invitrogen, Carlsbad, CA, USA), and quality and quantity were assessed spectrophotometrically before the cDNA was synthesized using the Thermo Scientific RevertAid Reverse Transcriptase in a Bio-Rad My Cycler™ Termal Cycler.

An ABI 7500 Fast Real-Time PCR system was used to perform the qRT-PCR (Applied Biosystems, Foster City, CA, USA) with the SYBR Green Real-time Master Mix. The *LOX1* and *LOX2* primer sequences in this study are detailed in Table 1. The housekeeping

soybean elongation factor (*ELF1b*) was used as a reference gene for relative quantification and the target expression relative to the housekeeping was used for ANOVA.

## Isoflavonoid determination

Isoflavonoid accumulation was determined after 72 h of herbivory treatment in seeds at R6 according to *Fehr et al. (1971)*, the lipid phase of seeds was removed with cyclohexane (*Zavala et al., 2015*). A total of 0.2 g of pulverized seeds were placed in a 50 ml tube and mixed with 10 mL of cyclohexane and incubated for 6 h. Then, tubes were centrifuged at 1,500 rpm for 10 min. Supernatant was discarded and the pellet was used for the extraction step. Isoflavonoids were extracted with methanol plus rutin that was used as an extraction internal control (0.1 g/5 ml MeOH). A total of 10 mL of methanol were added to 50 mL tubes and incubated for 6 h at room temperature. Then tubes were centrifuged at 1,500 rpm for 10 min. Supernatant was saved. The solvent was then evaporated at 40 °C and samples were re-dissolved with 500 μL of MeOH. Samples were purified with a C18 Silica Cartridge (Agilent Technologies, Palo Alto, Santa Clara, CA, USA) using different mixtures of MeOH–$H_2O$. Aliquots of 5 μL were subjected to high-performance liquid chromatography (HPLC; Agilent 1100 A series, Waldbronn, Germany) using a reverse-phase octadesyl column (Eclipse XDB-C18 4.6 × 150 mm, 5 μm). Isoflavonoids were eluted using a mobile phase gradient of 15–60% acetonitrile in 0.1% acetic acid for 60 min at a flow rate of 1 mL min$^{-1}$. Compounds were measured with the detector set at λ 270 nm. Retention times and quantitative data for daidzin, daidzein, genistin and genistein were obtained by comparison to known standards (all from Sigma–Aldrich, St. Louis, MO, USA). External standard curves were performed for isoflavonoid quantification.

## Statistical analysis

The experiments were set up as a completely randomized design with factorial treatments. One factor was the genotype (IAC-100 and Davis) and the second was insect damage (control and treated). Data were analyzed with the Infostat v. 2011 statistical package (*Di Rienzo et al., 2011*). Hardness analysis of pods, and SOD, CAT and GPOX activities as well as seed isoflavonoid contents in each experiment were analyzed using analysis of variance (ANOVA). Data on isoflavonoid content were transformed as needed to meet the assumptions of ANOVA by square root transformation. We applied the Tukey pairwise multiple comparisons procedure for studying GENOTYPE and TREATMENT effects.

Hierarchical clustering of oxidative stress enzymes was performed with the same data used for ANOVA using Euclidean distances in Infostat.

## RESULTS

### Damage and hardness assessment of pods

Oxidative damage caused by stink bugs was observed by visual analyses of $H_2O_2$ generation. Davis seeds affected by *Nezara viridula* exhibited highly enhanced brownish

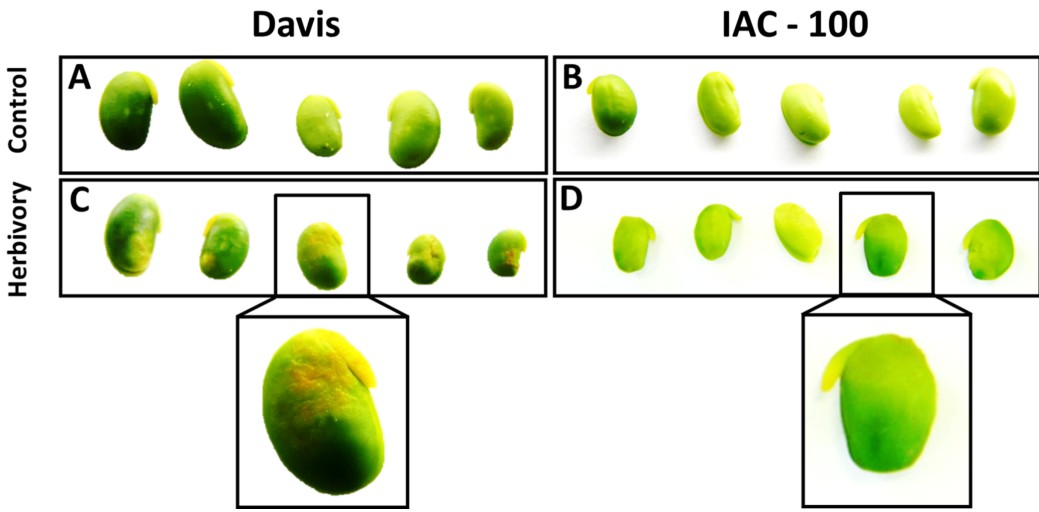

**Figure 1 H₂O₂ generation in sites damaged by stink bug.** Brown color shows a more affected condition. (A) Cultivar Davis with control treatment. (B) Cultivar IAC-100 with control treatment. (C) Cultivar Davis with herbivory treatment. (D) Cultivar IAC-100 with herbivory treatment. Images were acquired with a digital camera Nikon Coolpix L110.

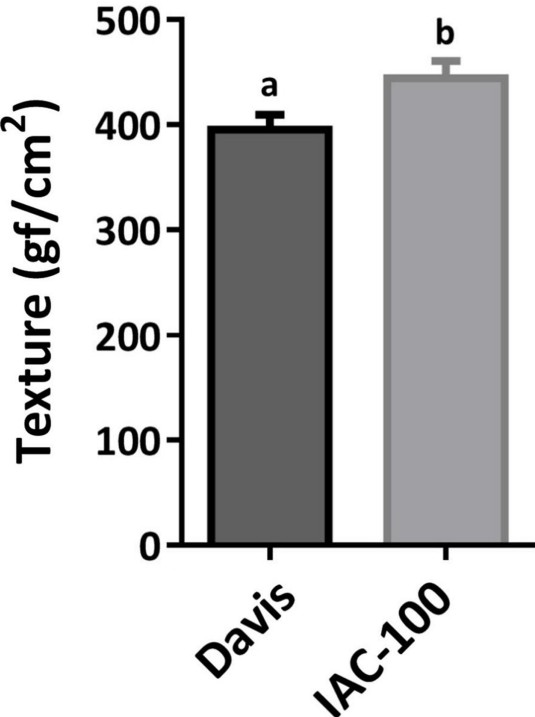

**Figure 2 Mean hardness of soybean pods in R6.** Means values (bars) and SEM (whiskers) are shown. Different letters are significantly different ($p < 0.05$, $n = 17$). Data source in Supplemental Information.

staining compared to the IAC-100 genotype (Fig. 1). Textural analysis of pods showed that IAC-100 has greater resistance to penetration than Davis ($p < 0.05$; Fig. 2). This can be a physical barrier to the insect's stylet penetration.

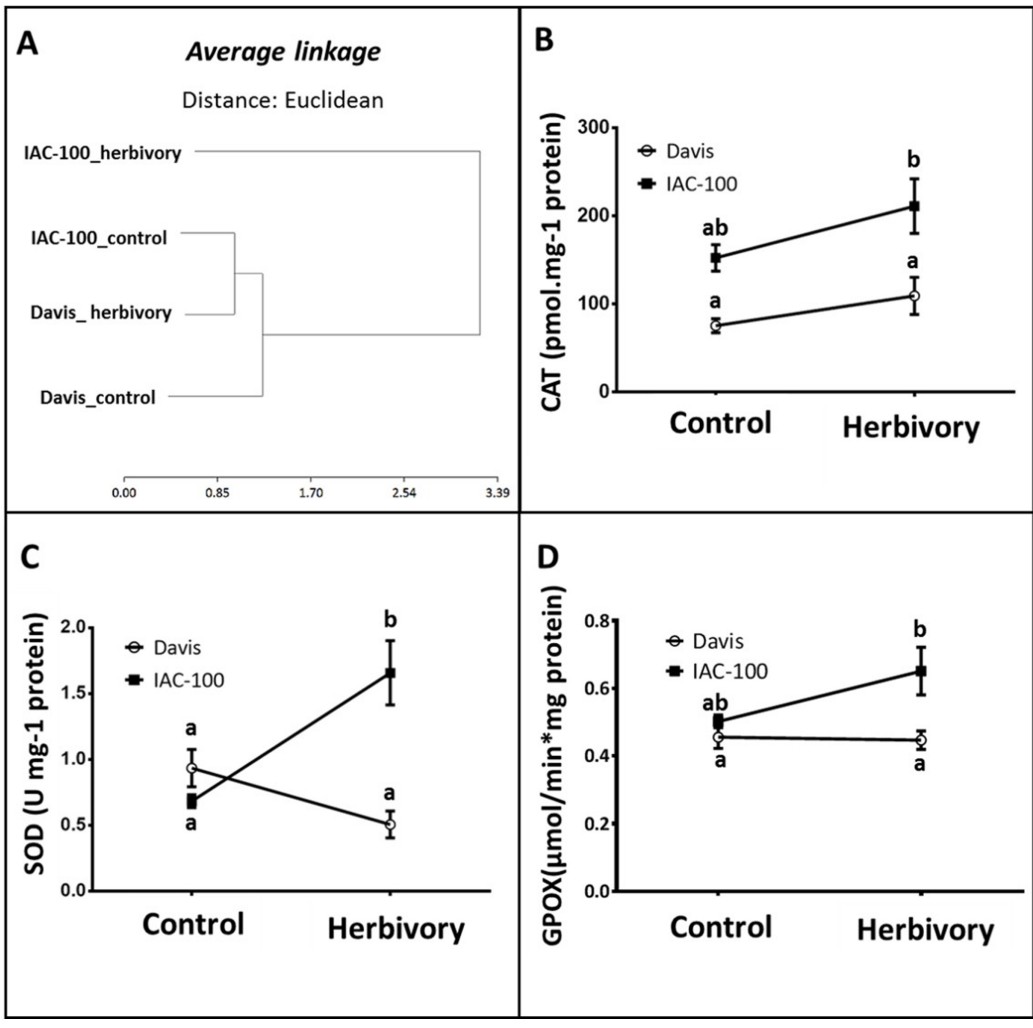

**Figure 3 Oxidative stress analysis in seeds of different soybean genotypes damaged by *Nezara viridula* and their respective controls without herbivory.** (A) Cluster analysis of oxidative stress enzymes. (B) Catalase activity. (C) Superoxide dismutase activity. (D) Guaiacol peroxidase activity. Means values (dots and squares) and SEM (whiskers) are shown. Different letters are significantly different ($p < 0.05$). Data source in Supplemental Information.

## Stress metabolism

According to our results, the cluster analysis of oxidative stress enzyme activities (CAT, SOD, GPOX) that was carried out using the statistical package Infostat v. 2011 (*Di Rienzo et al., 2011*) showed two main clusters that grouped the genotypes with the treatments. Figure 3A shows one cluster containing IAC-100 treated with stink bugs, and the second cluster containing IAC-100 control, Davis treated and Davis control. When exposed to insect treatment, the cluster distribution suggests a differential induction of enzyme activities between the resistant and susceptible genotypes. In addition, there is no Genotype x Treatment interaction in CAT activity and the differences are due to Genotype and Treatment ($p < 0.05$; Fig. 3B). Although the CAT enzyme is induced in both genotypes,

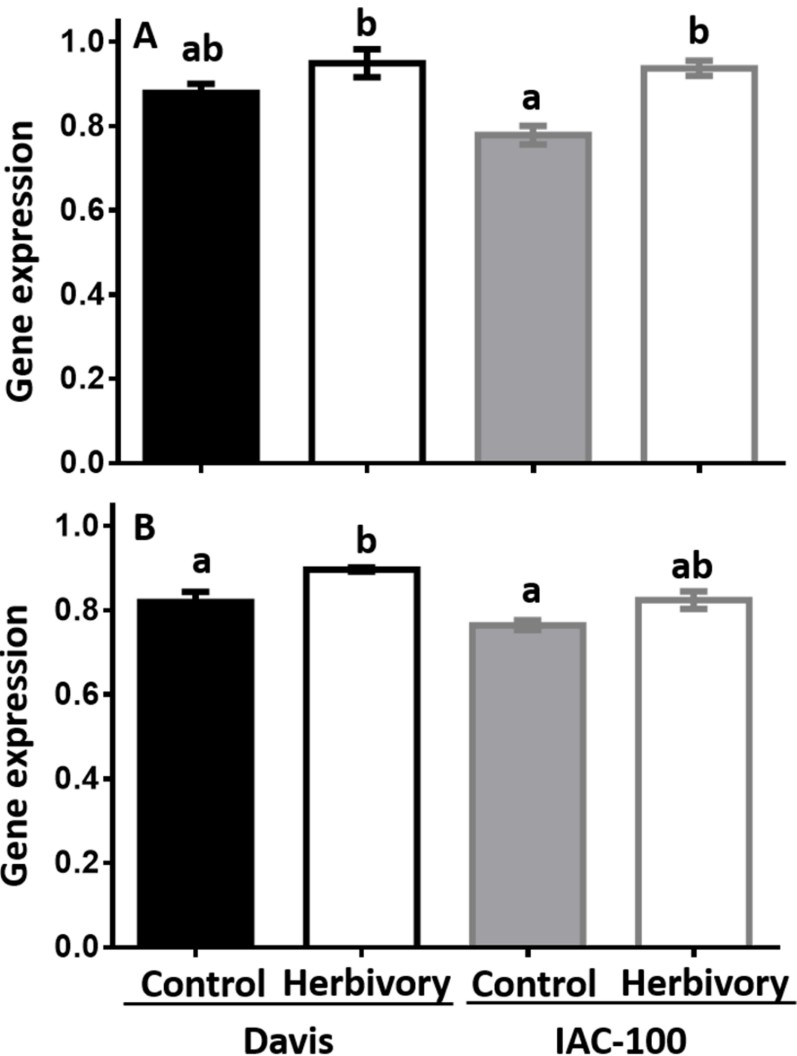

**Figure 4 Gene expression by Real-Time PCR of LOX1 and LOX2 in soybean seeds, either undamaged and damaged by *Nezara viridula*.** (A) LOX1. (B) LOX2. Gene expression is relative to the soybean elongation factor (*ELF1b*). Means values (bars) and SEM (whiskers) are shown. Different letters are significantly different ($p < 0.05$). Data source in Supplemental Information.

its activity is always higher in IAC-100 than in Davis. SOD activity was significantly greater in IAC-100 ($p < 0.05$) when affected by stink bugs (Fig. 3C). When comparing cultivars, GPOX showed a higher activity in IAC-100 both with and without damage caused by the insect ($p < 0.05$, Fig. 3D).

## Lipoxygenase expression quantification

Expression of *LOX1* and *LOX2* was induced by herbivory in both soybean cultivars when compared to their respective controls (Fig. 4). It suggests that soybean plants respond to this type of stress by increasing *LOX* expression after 24 h following treatment ($p < 0.05$).

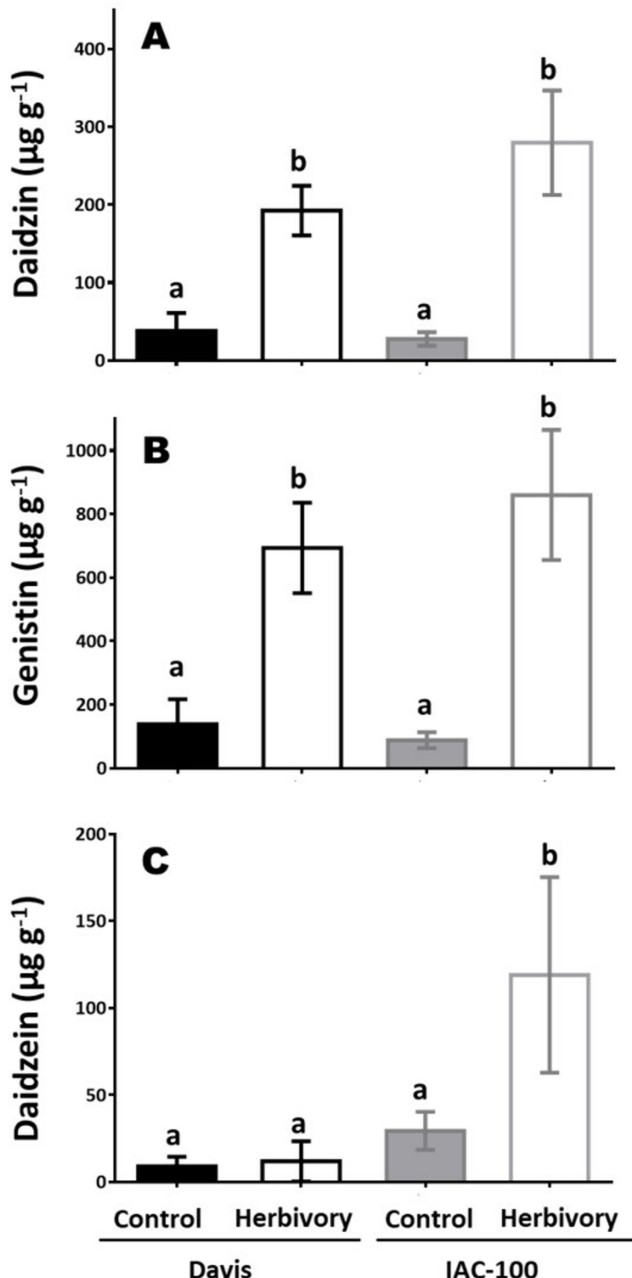

**Figure 5  Isoflavonoid concentration in seeds of cultivars Davis and IAC-100 damaged by stink bugs and their respective controls without herbivory.** (A) Daidzin concentration. (B) Genistin concentration. (C) Daidzein concentration. Means values (bars) and SEM (whiskers) are shown. Different letters are significantly different ($p < 0.05$). Data source in Supplemental Information.

## Isoflavonoids

Concentrations of isoflavonoids in soybean seeds showed differences between cultivars and types of compounds (Fig. 5). After herbivory, the concentration of daidzin and genistin increased in both genotypes and constitutive concentrations were similar and very low ($p < 0.05$; Figs. 5A and 5C). Higher levels of daidzein were found in the IAC-100 genotype,

regardless of the treatment, with a strong uprising trend under herbivory ($p < 0.05$; Fig. 5B). Except for one repeat of the cultivar Davis genistein was not detected in any treatment (Supplemental Information).

## DISCUSSION

Previous experiments with the IAC-100 soybean genotype reported low incidence of stink bugs (*McPherson, Buss & Roberts, 2007*). This cultivar has PI 229358 (*Clement & Quisenberry, 1998*) and PI 274454 (*Souza et al., 2015*) in its genealogy (*Carrao-Panizzi & Kitamura, 1995*), which exhibits a high degree of resistance to the stink bug complex. Some studies have shown antixenosis in IAC-100 against *N. viridula*, *P. guildinii* and *E. heros* (*Campos et al., 2010*; *Silva et al., 2014*; *De Souza et al., 2014*), and biological effects through isoflavonoids on *N. viridula* (*Piubelli et al., 2003a*). Our study suggests that cultivar IAC-100 maybe more tolerant to the southern green stinkbug due to greater hardness in pods, which serves as the first line of defense. In addition, a higher antioxidative enzyme activity was involved in plant defense, and it showed high levels of isoflavonoids in response to insect attack which, according to some authors (*Lane et al., 1985*; *Murakami et al., 2014*; *Simmonds & Stevenson, 2001*) affect the performance and survival of the herbivores.

Antixenosis in resistant plants can be related to morphological factors such as thickened plant epidermal layers, waxy deposits on leaves, stems, or fruits, nutritional deficiency, and chemical compounds (*Smith, 2005*). The thickness of various plant tissues influences the degree of resistance in some crop cultivars. Stink bug (*N. viridula*) damage elicits activation of MAPK signal in soybean seeds and induced salicylic acid that induced genes related with cell wall restructuration and increased lignin content, which can increase resistance to new insect attack by hardening cell walls (*Giacometti et al., 2018*). In this study, pods of IAC-100 cultivar showed greater resistance to penetration when compared to those of Davis, suggesting that hardening may be a physical barrier to insect feeding (Fig. 2).

Under stress conditions in plants, such as insect feeding, many signaling pathways are activated by different types of ROS causing "oxidative burst" (*Maffei, Mithöfer & Boland, 2007*). After insect attack, ROS are accumulated acting as the first barrier against the attack of pathogens and herbivores. However, to avoid self-toxicity by ROS, plant cells have ROS scavenging systems that help to remove the excess concentration and maintain a relatively low and constant one (*War et al., 2012*). Tolerance in plants is generally related to the control capacity of the cellular redox state resulting in lower oxidative damage (*Slesak et al., 2007*). ROS accumulation was detected after the feeding period of the chewing caterpillar *S. littoralis* or the piercing-sucking action of the spider mite *Tetranychus* urticae (*Maffei et al., 2006*; *Afify et al., 2011*). Infestation of lima beans by *S. littoralis* revealed, trough diaminobenzidine (DAB) staining, accumulation of $H_2O_2$ around the wounded area (*Maffei et al., 2006*). In our study we found that IAC-100 had less accumulation of $H_2O_2$ after insect feeding when compared to the susceptible genotype, which suggests that it has a higher capacity to control the redox state. A previous study demonstrated that ROS detoxification enzymes were only induced in $H_2O_2$-accumulating

resistant lines (*Moloi & Van der Westhuizen, 2008*). POD is one of such group of antioxidative enzymes, which scavenges the ROS besides having other defensive roles because they are also an important component of the immediate response of plants to insect damage (*War et al., 2012*). Catalases remove the $H_2O_2$ and reduce $H_2O_2$ to $2H_2O$. And SOD constitutes a frontline in the defense against ROS as they catalyze the dismutation of $O_{2-}$ (superoxide radical) to $H_2O_2$ (*Caverzan, Casassola & Brammer, 2016*). In wheat, an increase in *SOD* transcript in response to differential heat shock treatment was related to an enhanced tolerance to environmental stresses (*Kumar et al., 2013*). The present study demonstrates that the resistant cultivar had more control of the cellular redox state given its high level of enzyme activity after herbivory in SOD, GPOX y CAT (Fig. 3).

Many studies have shown that LOXs are implicated in the defense systems of several plant species against pathogens and insects (*Koch et al., 1992*; *Melan et al., 1993*; *Felton et al., 1994*; *Christensen et al., 2013*). Induction of LOX activity after herbivory has previously been studied in tomato in response to *Helicoverpa armigera* (*Yan et al., 2013*), in Arabidopsis following infestation by *Myzus persicae* and in lima bean in response to *Tetranychus urticae* (Moran & Thompson, 2001; *Arimura et al., 2000)*. In developing seeds of soybean, we found an induction of *LOX1* and *LOX2* after herbivory in both genotypes under study suggesting that this pathway is activated in response to the damage caused by *Nezara viridula*.

LOX activity regulates the production of the hormone JA, which modulates flavonoids and isoflavonoids that protect plants against insect pests and affect the behavior, growth and development of insects (*Simmonds, 2003*). In Arabidopsis, resistance against *Spodoptera frugiperda* is enhanced overexpressing a transcription factor that controls flavonoid production (*Johnson & Dowd, 2004*). In soybean leaves, flavonoids negatively affected the behavior of *Aphis glycines* and *Trichoplusia ni* (*Meng et al., 2011*; *Neupane & Norris, 1990*). Moreover, soybean seed damage caused by soybean pod borer (*Leguminivora glycinivorella*) was negatively correlated with higher levels of isoflavonoids, and showed positive correlations between isoflavonoid content in pods and undamaged seeds in treatments with stink bugs (*Nezara viridula* and *Piezodorus guildinii*) (*Zhao et al., 2015*; *Zavala et al., 2015*). Our results reveal that stink bug herbivory differentially promoted isoflavonoid production between genotypes with variations according to the type of compound. It can thus be concluded that not only total production of isoflavonoids but also the type of isoflavonoid produced can affect insect behavior.

## CONCLUSIONS

Taken the results together, it can be inferred that stink bug herbivory injury generates cell wall changes, induces pathways related to oxidative stress and secondary metabolites in developing seeds of soybean. As a result of the resistance characteristics to insects, such as control of cellular redox state and production of secondary metabolites, IAC-100 is a promising cultivar for a breeding program. However, further studies are still required to understand the functions of genes and the regulatory factors involved in defense responses in order to develop molecular markers for a breeding program.

The study of host plant resistance is an important alternative for producers in Argentina, but even more in Brazil, not only because of the high insect pressure but also for the phytopathogenic fungi that lead to the frequent use of chemical control.

## ACKNOWLEDGEMENTS

GDM Seeds for the availability for the use of greenhouses and other infrastructure. Verónica Feuring and Alina Crelier for technical assistance. Romina Giacometti and Natalia Ilina for the help in molecular biology methods.

### Funding

This work was supported by the Universidad de Buenos Aires (UBACYT 20020170100506BA – 2018) and the Agencia Nacional de Promoción Científica y Tecnológica (PICT 2685/2015). The funders had no role in study design, data collection and analysis, decision to publish, or preparation of the manuscript.

### Grant Disclosures

The following grant information was disclosed by the authors:
Universidad de Buenos Aires: UBACYT 20020170100506BA – 2018.
Agencia Nacional de Promoción Científica y Tecnológica: PICT 2685/2015.

### Competing Interests

The authors declare that they have no competing interests. Ivana Sabljic is currently a GDM employee but at the time this research was conducted she was a PhD student and a CONICET fellow.

### Author Contributions

- Ivana Sabljic conceived and designed the experiments, performed the experiments, analyzed the data, prepared figures and/or tables, authored or reviewed drafts of the paper, and approved the final draft.
- Jesica A. Barneto performed the experiments, analyzed the data, prepared figures and/or tables, authored or reviewed drafts of the paper, and approved the final draft.
- Karina B. Balestrasse analyzed the data, prepared figures and/or tables, authored or reviewed drafts of the paper, and approved the final draft.
- Jorge A. Zavala analyzed the data, prepared figures and/or tables, authored or reviewed drafts of the paper, and approved the final draft.
- Eduardo A. Pagano conceived and designed the experiments, analyzed the data, prepared figures and/or tables, authored or reviewed drafts of the paper, economical supporting, and approved the final draft.

### Data Availability

The raw measurements are available in the Supplemental File.

## Supplemental Information

Supplemental information for this article can be found online at http://dx.doi.org/10.7717/peerj.9956#supplemental-information.

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
