# Peer review of "Role of reactive oxygen species and isoflavonoids in soybean resistance to the attack of the southern green stink bug"

_PeerJ, doi:10.7717/peerj.9956_

## Round 0.1 · original submission · Minor Revisions

Dear authors, we thank you for your contribution to the management of knowledge on the subject Role of reactive oxygen species and isoflavonoids in soybean resistance to the attack of the southern green stink bug. Our decision is publishable with minor revisions. So, to improve the quality of your work, we attach the comments of the reviewers and suggest the following observations:
line between 43-51: add importance of biological controllers use
line 119: specific the measure of diet, where you mention the components.

Thanks for your contributions.

Carlos

Reviewer 1 ·

Basic reporting

The information contained in this section is quite good and I adore the subject given that you can see that they have been working on the subject.

Experimental design

I am left in doubt because two batches of seeds are referred to one to determine the response to oxidative stress in which 128 seeds are used and an adult is mentioned. Is this was introduced.? Is it enough for analysis?

reading line 125, I am left in doubt when you write when

Validity of the findings

Very good, according to the importance of soy in that country and in all where it is considered an important agricultural activity.

It should be borne in mind that these analyze are specific for specific sites, since the genotype plus the environment give the characteristic of both plaice and insects.

Additional comments

Very good information and findings, but it is specific to specific sites.
Maybe other varieties show gold behavior pattern

Reviewer 2 ·

Basic reporting

The language and narrative are well done.,

it has been used relevant literature review explaining the issue and research done. Regarding the background, it will be important to refer to the context geographically because the production system and crop are context-dependent.

The structure follows the PeerJ standards, discipline norms and portrays clarity

The figures are suitable for the report; however, they require to show the source i.e. to show where the data has been created or taken from. Fig1 and Fig5 require the source.


Raw data are suitable. Nevertheless, it requires to write the sources in each figure in the supplemental file raw data

Language is clear and properly used

Experimental design

The manuscript is original within the scope of the journal


The research question is not explicitly stated. the knowledge gap has been identified


The investigation follows the technical and ethical standards.


The method can be replicated and it has been well defined in the narrative

Validity of the findings

The benefit of the literature is clearly stated. Replication is encouraged to be taken.

The data is robust, statistically well defined, and controlled.


There is not speculation stated in the manuscript

The inferred conclusion has not relation to a research question due to the lack of research question in the manuscript

Additional comments

It would be interesting to refer to the implications of the research for the Latin American context. Including stating what type of producers can access this technology once it is available.

Reviewer 3 ·

Basic reporting

No comments

Experimental design

The methodology used for analyses performing is reasonably adequate, but I would to issue some questions and address some suggestions to the authors:
Firstly, I would like to know the reason of choosing Davis as the susceptible control. This is an old material and much less productive than the current cultivars used by farmers, which turn the comparison rather unrealistic to reach the manuscript objective of indicating IAC-100, as a source of resistance. The comparison of a resistant with a contemporaneous cultivar would be more realistic and useful for soybean breeders.
One other question is concernning to soybean stage. Why R6 (fully grown seed) was used for performing the analysis?
On the isoflavone detection, I have further comments. Soybean inmmature grains/seeds, in general, present three isoflavones (daidzin, genistin and glycitein) in two conjugates form (glucosyl and malonyl), and three aglycones (daidzein, genistein, and glycitein).
‘IAC 100’ has been the object of some studies about secondary metabolite and its chemical profile concerning flavonoids is relatively well known. There is a paper published by Graça et al, em 2016, (Phytochemistry 131: 84-91), where it is possible to see the isoflavone derivatives detected in IAC 100 extracts. In this paper, the authors detected and quantified the possible isoflavones forms (glycosides and aglycones) in IAC 100 seed extracts detectabled by HPLC analyses (as here).
When checking chromatograms provided in this manuscript, it is possible to see that not all compound (peaks) were identified. Thus, my suggestion for further studies is to detect the other isoflavone conjugates, as they inform the balance of such compounds. All isoflavone derivative standards are available in the market.

Validity of the findings

The manuscript relates a series of analysis to identify and quantify physical and biochemical defense mechanisms soybean plants to the Nezara viridula attack. The authors compared the response of cultivars, IAC-100 (insect resistant) with Davis (insect susceptible). In general, the manuscript is well written, and the results are interesting and represent a good contribution to clarify the mechanisms of soybean resistance to one important soybean insect-pest. The methodology used for analyses performing is reasonably adequate, results are interesting, and the discussion is appropriate.

Additional comments

In general, the manuscript is well written, and the results are interesting, representing a good contribution to clarify the mechanisms of soybean resistance to one important soybean insect-pest. The methodology used for analyses performing is reasonably adequate, results are interesting, and the discussion is appropriate.
In Introduction, line 44, please, add Euschistus heros, besides of Nezara viridula and Piezodorus guildini, as one of the predominant specie of stink bug in South America. Brazil is a big soybean producer country, and the more abundant and damaging stink bug in its soybean cropping area is the brown stink bug, E. heros. The other two species are main pests in farms located in the Brazilian States closer to Argentina and Uruguay.
After few clarifications, the manuscript is adequate to be accepted to PeerJ publication.

---

## Round 0.2 · accepted · Accept

Dear author,

I have reviewed your improvements (minor revisions), which were mostly more technical clarifications, therefore as Editor, I accept your contribution.